# Hierarchical Quantized Autoencoders

**Will Williams**[*]
willw@speechmatics.com

**Sam Ringer**[*]
samr@speechmatics.com

**John Hughes**
johnh@speechmatics.com

**Tom Ash**
toma@speechmatics.com

**David MacLeod**
davidma@speechmatics.com

**Jamie Dougherty**
jamied@speechmatics.com

## Abstract

Despite progress in training neural networks for lossy image compression, current approaches fail to maintain both perceptual quality and abstract features at very low bitrates. Encouraged by recent success in learning discrete representations with Vector Quantized Variational Autoencoders (VQ-VAEs), we motivate the use of a hierarchy of VQ-VAEs to attain high factors of compression. We show that the combination of stochastic quantization and hierarchical latent structure aids likelihood-based image compression. This leads us to introduce a novel objective for training hierarchical VQ-VAEs. Our resulting scheme produces a Markovian series of latent variables that reconstruct images of high-perceptual quality which retain semantically meaningful features. We provide qualitative and quantitative evaluations on the CelebA and MNIST datasets.

## 1 Introduction

The internet age relies on lossy compression algorithms that transmit information at low bitrates. These algorithms are typically analysed through the rate-distortion trade-off, originally posited by Shannon [33]. When performing lossy compression at extremely low bit rates, obtaining low distortions often results in reconstructions of very low perceptual quality [5, 6, 38]. For modern lossy compression, high perceptual quality of reconstructions is often more desirable than low distortions. This work investigates good performance on this rate-perception tradeoff as opposed to more standard rate-distortion trade offs, with a focus on the low-rate regime.

At low bitrates it is desirable to communicate only high-level concepts and offload the 'filling in' of details to a powerful decoder [38]. Neural Networks present a promising avenue since they are flexible enough to learn the complex transformations required to both capture such high-level concepts and reconstruct in a convincing way that avoids artifacts [32, 10, 14].

Variational Autoencoders (VAEs [15]) are latent variable Neural Network models that have made significant strides in lossy image compression [35, 1]. However, due to a combination of a poor likelihood function and a sub-optimal variational posterior [31, 43], reconstructions can look blurred and unrealistic [44, 11]. There have been many attempts to construct hierarchical forms of both VAEs and Vector Quantized Variational Autoencoders (VQ-VAEs), however perceptual quality is frequently sacrificed at low-rates, and has only recently been made viable with methods that require large autoregressive decoders [8, 30]. Solutions to this problem then take two forms: either augmenting the likelihood model, for instance, by using adversarial methods [38] or improving the structure of the posterior/latent space [43, 3]. However, at low rates both solutions struggle to match the realism of implicit generative models [9].

---

[*]Equal contribution.

Table 1: CelebA interpolations of the HQA encoder output $z_e$ in the 9 bit 8x8 latent space. The original 64x64 images are shown on the left and right. The center images are the resulting decodes when using 8 linearly interpolated points between the $z_e$ of the original images. Compression is from 98,304 to 576 bits (171x compression).

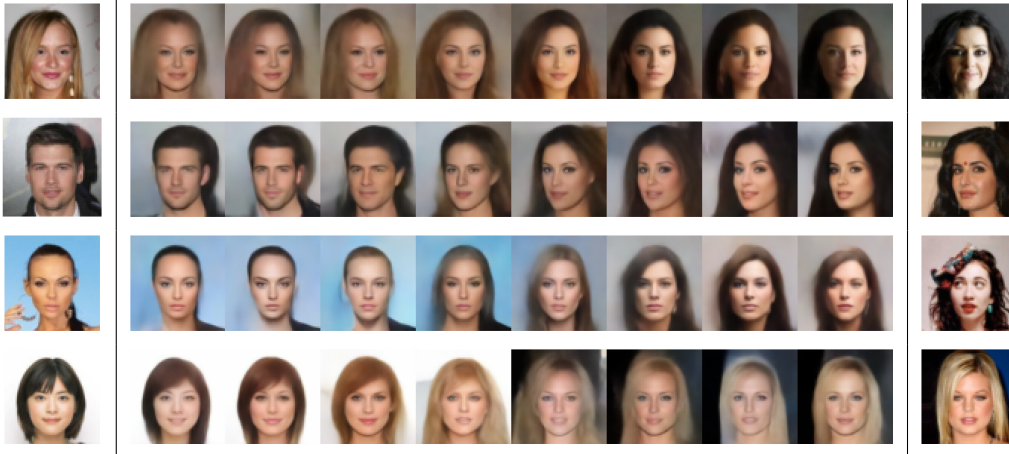

To address these issues, we build from previous work on heirarchical VQ-VAEs and introduce[1] the 'Hierarchical Quantized Autoencoder' (HQA). Our system implicitly gives rise to many of the qualities of explicit perceptual losses and furnishes the practitioner with a repeatable operation of learned-compression that can be trained greedily.

Our key contributions are as follows:

- We introduce new analysis as to why probabilistic quantized hierarchies are particularly well-suited to optimising the perception-rate tradeoff when performing extreme lossy compression.
- We propose a new scheme (HQA) for extreme lossy compression. HQA exploits probabilistic forms of VQ-VAE's commitment and codebook losses and uses a novel objective for training hierarchical VQ-VAEs. This objective leads to higher layers implicitly reconstructing the full posterior of the layer below, as opposed to samples from this posterior.
- We show that HQA can produce reconstructions of high perceptual quality at very low rates using only simple feedforward decoders, where as related methods require autoregressive decoders.

## 2 Related Work

### 2.1 Lossy Compression and the Rate-Perception Trade-off

Shannon's rate-distortion theory of lossy compression makes no claims about perceptual quality. Blau and Michaeli [6] show that optimising for distortion necessitates a trade off with perceptual quality, particularly at extremely low rates. This move to focus on perceptual quality has motivated the introduction of perceptual losses [36, 4, 32, 27] which are heuristically defined and attempt to capture different aspects of human-perceived perceptual quality. Our work naturally gives rise to losses at different levels of abstraction which have a similar effect as perceptual losses but which are less heuristically defined and encourage abstract semantic categories to be captured. This leads to good performance on the rate-perception task on which we focus.

Blau and Michaeli [6] extend lossy compression to allow for stochastic decodes. Prior work [38, 2] notes that to achieve good perceptual quality at extreme rates, stochastic decoders are essential. Stochasticity has previously been introduced in an ad-hoc manner by injecting a noise vector into the decoder alongside the code. This is the same strategy used by most conditional generative models. However, this artificial introduction of stochasticity is problematic as the decoder often learns to ignore the noise vector completely [45, 12]. HQA parameterizes distributions over codes at different

layers of abstraction, each of which can be sampled from in turn. This introduces stochasticity in a more natural and nonrestrictive manner.

## 2.2 VAE hierarchies

Our work is most closely related to Gregor et al. [10], where a VAE-based hierarchy is constructed in an attempt to capture increasingly abstract concepts. Similarly, we only need to transmit top-level latents of a hierarchical model for use as a lossy code. However, their scheme relies on expensive iterative computation to decode latents and they struggle empirically to maintain perceptual quality at low rates. They rely on iterative refinement to obtain sharpness whereas our scheme can obtain a sharp and credible reconstruction with a single computational pass through the network. Additionally, they can only transmit a subset of the higher levels in the hierarchy, whereas each layer in our hierarchy represents a fully independent lossy code which can be transmitted at a fixed rate.

VQ-VAE-2 [30] introduces a hierarchy of VQ-VAEs and is trained using a two stage procedure. During the first stage all VQ-VAEs are trained jointly under one objective. During the second stage, large autoregressive decoders are trained and replace the original decoders. Although introduced as a generative model, the system after each of these stages can potentially be used for lossy compression. After the first stage, the structure of VQ-VAE-2 is such that the latents from *all* layers are required for image reconstruction. Therefore, all latents must be transmitted to perform lossy compression, making low-rate compression near impossible. The system after the second stage of training is more suitable for lossy compression as only the highest level latents need transmitting. However, the new decoders then dominate the parameter count in the final model by several orders of magnitude and their autoregressive nature lead to computationally burdensome reconstruction times. Additionally, for each fixed compression rate, a whole new VQ-VAE-2 must be trained through both stages. Instead, we look to compare against schemes that use simple feedforward decoders and that have feasible scaling properties across many bitrates.

One such scheme is the Hierarchical Autoregressive Model [8] (denoted HAMs). Similar to VQ-VAE-2, HAMs train a hierarchy of VQ-VAEs in a two step procedure, with the second step training a series of autoregressive auxillary decoders. In contrast to VQ-VAE-2, the hierarchy obtained after the first stage is suitable for extreme lossy compression as only the top level latents need to be transmitted and only simple feedforward decoders are used. In contrast to HQA, each layer of HAMs produces a deterministic posterior and each decoder is trained with a cross-entropy loss over the code indices of the layer below.

# 3 Background

## 3.1 VQ-VAE

VQ-VAEs [39, 30] model high dimensional data $x$ with low-dimensional discrete latents $z$. A likelihood function $p_\theta(x|z)$ is parameterized with a decoder that maps from latent space to observation space. A uniform prior distribution $p(z)$ is defined over a discrete space of latent codes. As in the variational inference framework [25], an approximate posterior is defined over the latents:

$$q_\phi(z = k|x) = \begin{cases} 1 & \text{for } k = \text{argmin}_j ||z_e(x) - e_j||_2 \\ 0 & \text{otherwise} \end{cases} \tag{1}$$

The codebook $(e_i)_{i=1}^N$ enumerates a list of vectors and an encoder $z_e(x)$ maps into latent space. A vector quantization operation then maps the encoded observation to the nearest code. During training the encoder and decoder are trained jointly to minimize the loss:

$$- \log p_\theta(x|z = k) + ||sg[z_e(x)] - e_k||_2^2 + \beta ||z_e(x) - sg[e_k]||_2^2, \tag{2}$$

where *sg* is a stop gradient operator. The first term is referred to as the *reconstruction loss*, the second term is the *codebook loss* and the final term is the *commitment loss*. In practice, the codes $e_k$ are learnt via an online exponential moving average version of k-means.

# 4 Lossy Compression Using Quantized Hierarchies

Lossy compression schemes will invariably use some form of quantization to select codes for transmission. This section examines the behaviour of quantization-based models trained using maximum-likelihood.

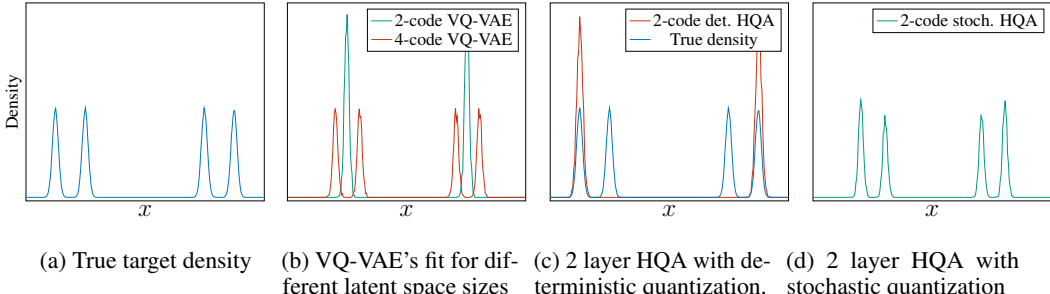

(a) True target density  (b) VQ-VAE's fit for different latent space sizes  (c) 2 layer HQA with deterministic quantization.  (d) 2 layer HQA with stochastic quantization

Figure 1: Modelling a simple multi-modal distribution using different forms of hierarchies. The HQA system uses the pre-trained 4-code VQ-VAE from Figure 1b and adds a 2-code VQ-VAE on top. Note, for HQA, only the top-layer codes count for transmission since the lower level codes are generated during decoding.

## 4.1 Illustrative task

Consider performing standard lossy compression on datapoints sampled from the distribution shown in Figure 1a. Each datapoint is encoded to an encoding consisting of only a small number of bits. Each encoding is then decoded to obtain an imperfect (lossy) reconstruction of the original datapoint. We desire a lossy compression system that shows the following behaviour:

**Low Bitrates** The encoding of each datapoint should consist of as few bits as possible.

**Realism** The reconstruction of each datapoint should *not* take on a value that has low probability under the original distribution. For the distribution in Figure 1a, this corresponds to regions outside of the four modes. We term such reconstructions *unrealistic*. In other words, it should never be the case that a reconstruction is *clearly* not from the original distribution. A link can be drawn between these areas of low probability in the original data distribution and the blurry/unrealistic samples often seen when using VAEs for reconstruction tasks.

## 4.2 Single Layer VQ-VAE

### 4.2.1 4-Code VQ-VAE

We begin by using a VQ-VAE to compress and reconstruct samples from the density shown in Figure 1a. We first train a VQ-VAE that uses a latent space of 4 codewords. The encodings produced by this VQ-VAE will therefore each be of size 2 bits ($= \log_2 4$). The red trace in Figure 1b shows the density of the reconstructions from this 4-code VQ-VAE. It is a perfect match of the density function that the original datapoints were sampled from. There are no *unrealistic* reconstructions as all reconstructed datapoints fall in regions of high density under the original distribution.

### 4.2.2 2-Code VQ-VAE

We now fit a VQ-VAE with a 2 codeword (1 bit) latent space to the original density. The green trace of Figure 1b shows the result. The mode-covering behaviour shown by this VQ-VAE causes reconstructions to fall in regions of low probability under the original distribution. Therefore, nearly all reconstructions are *unrealistic*. This mode-covering is a well known pathology of all likelihood-based models trained using the asymmetric divergence $KL[p_\theta(x)||p(x)]$. [22] show mode-covering limits the perceptual quality of reconstructions. To reiterate, this is because mode-covering produces *unrealistic* samples.

The question then arises: can we do better and produce *realistic* reconstructions using only a 1 bit encoding?

### 4.3 Quantized Hierarchies

We now take the pretrained 4-code VQ-VAE that produces the red trace in Figure 1b. We term this VQ-VAE Layer 1. We then train a *new* 2-code VQ-VAE, which we term Layer 2, to compress and reconstruct the encodings produced by Layer 1. The resulting system is a *quantized hierarchy*.

We can then compress and reconstruct datapoints sampled from the original distribution using the whole quantized hierarchy, as shown in Algorithm 1. Algorithm 1 is a simplification of the Hierarchical Quantized Autoencoder (HQA) described in Section 5.2.1.

---

**Algorithm 1** Lossy Compression Pseudo-code Using A Quantized Hierarchy

---

1: $x$: Datapoint to be compressed
2: $e_1 \leftarrow \text{Encoder}_1(x)$                                          ▷ Encode using Layer 1
3: $e_2 \leftarrow \text{Encoder}_2(e_1)$                                        ▷ Encode using Layer 2
4: $q_2 \leftarrow \text{Quantize}(e_2)$
5: Transmit $q_2$                                             ▷ $q_2$ is the final encoding
6: $\hat{e}_1 \leftarrow \text{Decoder}_2(q_2)$         ▷ Decode using Layer 2; this will mode-cover Layer 1's latent space
7: $\hat{q}_1 \leftarrow \text{Quantize}(\hat{e}_1)$                    ▷ Resolve mode-covering in the latent space
8: $\hat{x} \leftarrow \text{Decoder}_1(\hat{q}_1)$                                  ▷ Decode using Layer 1
9: $\hat{x}$: Lossy reconstruction of $x$

---

For VQ-VAE, each codeword is represented as a vector in a continuous latent space. If we consider the points in the latent space of Layer 1 that are actually used for encodings, there are only 4 points that are used: the locations of the 4 codewords. In other words, the distribution over the latent space of Layer 1 contains 4 modes.

Layer 2 is used to compress and reconstruct points from the latent space of Layer 1. As the latent space of Layer 1 contains 4 modes but Layer 2 only uses 2 codewords, Layer 2 will mode-cover for the same reasons described above. However, this mode-covering is now over the *latent* space of Layer 1 and not over the *input* space of the original distribution.

This mode-covering can now be resolved through quantization. The decode from Layer 2 can be quantized to a code in Layer 1's latent space (i.e quantized to a mode). Therefore, the reconstructions of Layer 1's latent space, and hence the final reconstruction, are more likely to be *realistic*.

VQ-VAE uses a deterministic quantization procedure which always quantizes to the code that is geometrically closest to the input embedding. We can use the quantized hierarchy introduced, along with deterministic quantization, to reconstruct samples from the original distribution. The result is shown by the red trace in Figure 1c. For the reasons outlined above, no mode-covering behaviour is observed and all reconstructions are *realistic*. However, mode-dropping is now occurring.

### 4.4 Stochastic Quantization

If a stochastic quantization scheme is introduced (c.f. Section 5.1) then this mode-dropping behaviour can also be resolved. Figure 1d shows the result of using the quantized hierarchy, now with stochastic quantization. No mode-dropping or mode-covering behaviour is present. Note that the quantized hierarchy uses 1 bit encodings, the same size as the encoding of the 2-code VQ-VAE that failed to model the distribution (c.f. Figure 1b). This result shows that, under a given information bottleneck, probabilistic quantized hierarchies allow for fundamentally different density modelling behaviour than equivalent single layer systems. Furthermore, unlike deterministic compression, there is no single decoded data; there are now many possible decodes.

Therefore, we propose that probabilistic quantized hierarchies can mitigate the *unrealistic* reconstructions produced by likelihood-based systems for the following reasons:

- **Hierarchy:** By choosing to model a distribution using a hierarchical latent space of increasingly compressed representations, mode-covering behaviour in the input space can be exchanged for mode-covering behaviour in the latent space. This also acts as a good meta-prior to match the hierarchical structure of natural data [17].
- **Quantization:** Quantization allows for the resolution of mode-covering behaviour in latent space, encouraging *realistic* reconstructions that fall in regions of high density in the input space.

- **Stochastic Quantization:** If quantization is performed deterministically then diversity of reconstructions is sacrificed. By quantizing stochastically, mode-dropping behaviour can be mitigated. In addition, this introduces the stochasticity typically required for low-rate lossy compression in a natural manner.

## 5 Method

### 5.1 Stochastic Posterior

We depart from the deterministic posterior of VQ-VAE and instead use the stochastic posterior introduced by Sønderby et al. [34]:

$$q(z = k|x) \propto \exp -||z_e(x) - e_k||_2^2 . \tag{3}$$

Quantization can then be performed by sampling from $q(z = k|x)$. At train-time, a differentiable sample can be obtained from this posterior using the Gumbel Softmax relaxtion [13, 24]. While training HQA, we linearly decay the Gumbel Softmax temperature to $0$ so the soft quantization operation closely resembles hard quantization, which is required when compressing to a fixed rate. At test-time we simply take a sample from Equation 3.

Crucially, under this formulation of the posterior, $z_e(x)$ (henceforth $z_e$) must be positioned well relative to *all* codes in the latent space, not just the nearest code [41]. As $z_e$ implicitly defines a distribution over all codes, it carries more information about $x$ than a single quantized latent sampled from $q(z|x)$. This is exploited by the HQA hierarchy, as discussed below.

### 5.2 Training Objective

#### 5.2.1 Single Layer

In a single layer model, the encoder generates a posterior $q = q(z|x)$ over the codes given by Equation 3. To calculate a reconstruction loss we sample from this posterior and decode. Additionally, we augment this with two loss terms that depend on $q$:

$$\mathcal{L} = \underbrace{- \log p(x|z = k)}_{\text{reconstruction loss}} - \underbrace{\mathcal{H}[q(z|x)]}_{\text{entropy}} + \underbrace{\mathbb{E}_{q(z|x)}||z_e(x) - e_z||_2^2}_{\text{probabilistic commitment loss}} . \tag{4}$$

This objective is the sum of the *reconstruction loss* as in a normal VQ-VAE (Equation 2), the *entropy* of $q$, and a term similar to the *codebook/commitment loss* in Equation 2 but instead taken over all codes, weighted by their probability under $q$. The objective $\mathcal{L}$ resembles placing a Gaussian Mixture Model (GMM) prior over the latent space and calculating the Evidence Lower BOund (ELBO), which we derive in Appendix B.

#### 5.2.2 Multiple Layers

When training higher layers of HQA, we take take the reconstruction target to be $z_e$ from the previous layer. This novel choice of reconstruction target is motivated by noting that the embedding of $z_e$ implicitly represents a distribution over codes. By training higher layers to minimize the MSE between $z_e$ from the layer below and an estimate $\hat{z}_e$, the higher layer learns to reconstruct a full distribution over code indices, not just a sample from this distribution. Empirically, the results in Section 6.2 show this leads to gains in reconstruction quality.

In this way, a higher level VQ-VAE can be thought of as reconstructing the full posterior of the layer below, as opposed to a sample from this posterior (as in Fauw et al. [8]). The predicted $\hat{z}_e$ is used to estimate the posterior of the layer below using Equation 4, from which we can easily sample to perform stochastic quantization, as motivated in Section 4.

The Markovian latent structure of HQA - where each latent space is independent given the previous layer - allows us to train each layer sequentially in a greedy manner as shown in Figure 2 (left). This leads to lower memory footprints and increased flexibility as we are able to ensure the performance of each layer before moving onto the next. Appendix D describes algorithm in full.

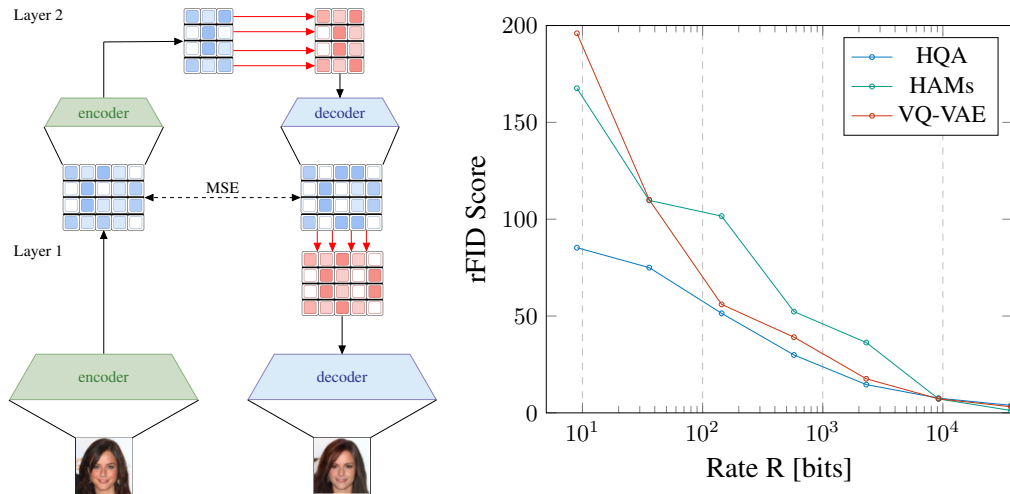

Figure 2: **Left**: System diagram of training the second layer of the HQA. Images are encoded into a continuous latent vector by Layer 1 before being encoded further by Layer 2. This representation is then quantized according to the stochastic posterior given by the red arrows, and then decoded by Layer 2. If training, an MSE loss is taken with this output and the input to the Layer 2 encoder. If performing a full reconstruction, the representation is quantized and then decoded by Layer 1. **Right**: Plot of rate against reconstruction FID (rFID) for compressing and reconstructing CelebA test examples.

## 5.3 Codebook Optimization

The loss given by Equation 4, in combination with the use of the Gumbel-Softmax, allows for the code embeddings to be learnt directly without resorting to moving average methods. This introduces a new pathology where codes that are assigned low probability under $q(z|x)$ for all $x$ receive low magnitude gradients and become unused. During training, we reinitialise these unused codes near codes of high usage. This results in significantly higher effective rates. Code resetting mirrors prior work in online GMM training [28, 40] and over-parameterized latent spaces [42].

## 6 Experiments

### 6.1 CelebA

To show the scalability of HQA and the compression rates it can achieve on natural images, we train on the CelebA dataset [21] at a 64x64 resolution. The resulting system is a 7-layer HQA, where the final latent space of 512 codes has size $1 \times 1$ due to downsampling by 2 at each layer. The architecture of each layer is detailed in Appendix C.

For comparison, we also train 7 different VQ-VAE systems. Each VQ-VAE has the same compression ratio and approximate parameter count as its HQA equivalent. We also compare against the hierarchical quantized system introduced by HAMs, since their system also can be used for low-rate compression with simple feedforward decoders (c.f. discussion in Section 2.2). As with the VQ-VAE baselines, each HAMs layer has the same compression ratio as its HQA equivalent. Table 2 shows reconstructions of two different images from the test set for each layer of HQA, as well as the reconstructions from the VQ-VAE and HAMs baselines.

Qualitatively, the HQA reconstructions display higher perceptual quality than both VQ-VAE and HAMs at all compression rates, with the difference becoming more exaggerated as the compression becomes more extreme. The high-level semantic features of the input image are also better preserved with HQA than with the baselines, even when the reconstructions are very different from the original in pixel space. For a quantitative comparison, we evaluate the test set reconstruction Fréchlet Inception Distance (rFID) for each system. Figure 2 (right) shows that HQA achieves better rFIDs than both VQ-VAE and HAMs and, as with the qualitative comparison, the difference becomes more exaggerated at low rates. We note the well known issues with relative comparison between

Table 2: Reconstructed CelebA test-set images at different levels of compression, with number of transmitted bits

| System | Original 98,304 | 2.7x 36,864 | 11x 9,216 | 43x 2,304 | 171x 576 | 683x 144 | 2,731x 36 | 10,923x 9 bits |
|---|---|---|---|---|---|---|---|---|
| HQA | | | | | | | | |
| HAMs | | | | | | | | |
| VQ-VAE | | | | | | | | |
| HQA | | | | | | | | |
| HAMs | | | | | | | | |
| VQ-VAE | | | | | | | | |

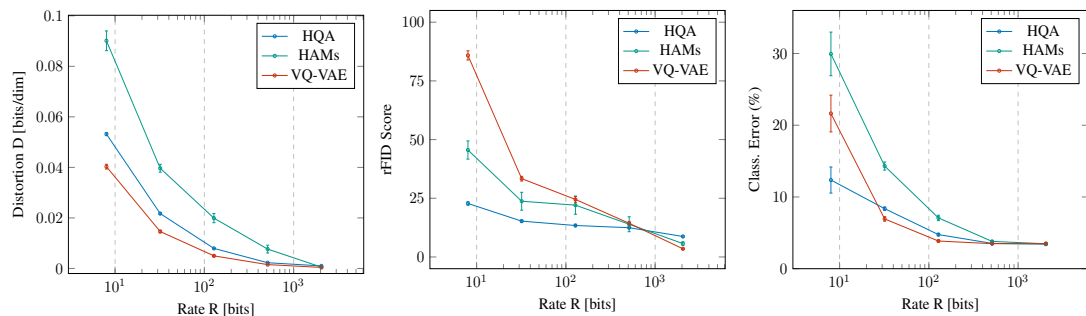

likelihood-based models and adversarially trained models when using rFID [29], and therefore only look to compare HQA with likelihood-based baselines.

## 6.2 MNIST

We performed an ablation study on MNIST [18] with the data rescaled to 32x32. In addition to measuring distortion and rFID, we evaluated how well each system was preserving the semantic content of each image by using a pre-trained MNIST classifier to classify the resulting reconstructions.

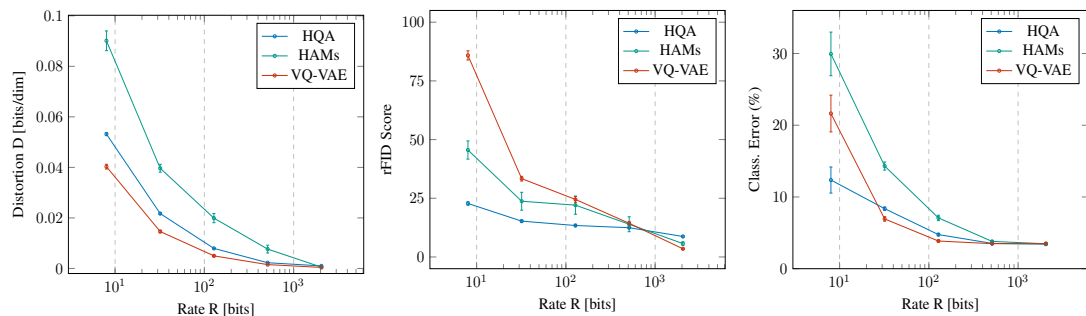

Figure 3: Plots of rate against distortion, reconstruction FID (rFID) and classification error for compressing and then reconstructing MNIST test examples. Error bars are 95% confidence intervals based on 10 runs with different training seeds.

We trained five layers, each compressing the original images by a factor of 2 in each dimension, such that the final layer compressed to a latent space of size 1x1. For VQ-VAE we trained to a 1x1 latent space directly. We control for the number of parameters (∼1M) in each system, training each with codebook size 256 and dimension 64.

Table 3 and Figure 3 both show that HQA has superior rate-perception performance (as approximated by rFID) at low rates than the other baselines. The trade-off between rate-perception and rate-distortion performance described by Blau and Michaeli [6] is clearly visible, resulting in HQA displaying worse distortions but better rFID scores. Furthermore, the classification accuracy results show that, at extreme rates, HQA maintains more semantic content from the originals when compared to the other methods.

Table 3: Distortion (MSE), reconstruction FID (rFID) and Classification Error scores for ablated systems, after compressing MNIST 10k test samples into an 8-bit 1x1 latent space then reconstructing. 'GS' covers introducing Gumbel Softmax and code resetting. 'MSE' means using Mean Squared Error loss on all layers. Errors represent a 95% confidence interval based on 10 runs.

| System | Distortion ↓ | rFID Score ↓ | Class. Error (%) ↓ | Reconstructions |
|---|---|---|---|---|
| No Compression | $0.000 \pm 0.000$ | $0.0 \pm 0.0$ | $3.13 \pm 0.00$ | 74593 |
| VQ-VAE | $\mathbf{0.040 \pm 0.001}$ | $85.9 \pm 2.0$ | $21.6 \pm 2.56$ | 79593 |
| + hierarchy (HAMs) | $0.090 \pm 0.004$ | $45.6 \pm 3.9$ | $29.9 \pm 3.04$ | 74593 |
| HAMs + GS | $0.108 \pm 0.009$ | $38.6 \pm 3.1$ | $51.1 \pm 6.28$ | 74392 |
| HAMs + MSE | $0.052 \pm 0.0004$ | $36.0 \pm 1.0$ | $11.1 \pm 0.46$ | 74593 |
| HAMs + GS + MSE | $0.054 \pm 0.0003$ | $\mathbf{21.0 \pm 1.0}$ | $\mathbf{10.6 \pm 0.93}$ | 74593 |
| + probabilistic loss (**HQA**) | $0.053 \pm 0.0006$ | $22.8 \pm 0.7$ | $12.4 \pm 1.82$ | 74593 |

Furthermore, the ablation study in Table 3 shows that, although the Gumbel-Softmax (GS) and MSE loss show improved performance when used individually, it is the combination of both that leads to the largest gain in performance, suggesting the benefits are orthogonal. Notably, HQA is the only system to give both good rFID and classification scores across all rates, the largest difference being at extreme compression rates. We note that the probabilistic loss of HQA hinders performance under the MNIST task. However, we empirically found that the probabilistic loss was essential to ensure stability of HQA when training on more complex datasets such as CelebA.

Table 4: Linear interpolations of encoder output $z_e$ in the 8 bit 1x1 latent space. The far left and right images are originals. Others are decoded from the interpolated quantized encoder output $z_q$.

Linear interpolations in Table 4 show that HQA has more dense support for coherent representations across its latent space than HAMs or VQ-VAE. Intermediate images for HQA are sharp and crisply represent digits, never deforming into unrealistic shapes. The same behaviour is observed for faces in the CelebA dataset, as shown in Table 1. Additional results can be found in Appendix A.

# 7 Conclusion

In this work, we introduce the 'Hierarchical Quantized Autoencoders', a promising method for training hierarchical VQ-VAEs under low-rate lossy compression. HQA introduces a new objective and is a naturally stochastic system. By incorporating a variety of additional improvements, we show HQA outperforms equivalent VQ-VAE architectures when reconstructing on the CelebA and MNIST datasets under extreme compression.

## Broader Impact

It is estimated that streaming of digital media accounts for 70% of today's internet traffic [19], and this is reflected by the increasing importance of high quality compact representations in the big visual data era [23]. Our research takes steps towards addressing this issue by providing a scalable architecture for semantically meaningful compression, at rates unachievable by traditional algorithms.

As well as the economic advantages of low-rate compression, there is the benefit of reduced energy and resources required for transmission and storage of smaller data, although this must be traded off against the currently higher computational cost of encoding/decoding.

Like most image based research, HQA has broader implications related to computer vision applications and the ethics surrounding them. As these are detailed by Lauronen [16] we instead choose to focus more directly on the potential consequences of our cited objective: to produce realistic and semantically consistent compressed images at low bitrates.

Whilst we observe empirically that the hierarchy of concepts retained by the HQA model can relate to a human idea of semantic importance, we do not control for this explicitly, which could have negative repercussions.

For example, in the case of human imagery it is possible for decoded characteristics related to ethnicity or gender to be misrepresentative of the original, a scenario which may be exacerbated by a biased training set. In a more general sense, it is possible that mission critical details could be removed or modified, and whilst this is symptomatic of all low bitrate lossy compressions schemes, the realism of the output could lead to an misguided interpretation which would traditionally be offset by the appearance of artifacts or a lower resolution output.

An interesting future research direction could be to alleviate this issue by conditioning the model on semantic labels as demonstrated by Agustsson et al. [2].

Further to this, the stochastic nature of our decodes means that the sender of an image has no way of knowing exactly what image the receiver will view and indeed different receivers of the same transmitted image will see different outputs. To a degree, viewers of media are used to this (for example where technologies automatically increase / reduce resolution according to available bandwidth), however methods such as ours have the potential to vary images in terms of higher level content as well as fine grained detail. This makes quality control, for example, problematic and use cases sensitive to this would need to do careful further investigation before using techniques such as ours. For other use cases however, such as artistic media, having a built in method for variable user experience may actually provide an interesting avenue for creative exploration.

## Footnotes

[1]Code available at `https://github.com/speechmatics/hqa`

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
