[Supplementary Material]

# A    Additional HQA Results

Table 5: Additional CelebA interpolations of the HQA encoder output $z_e$ in the 9 bit 8x8 latent space. Compression is from 98,304 to 576 bits (171x compression).

Table 6: CelebA reconstruction diversity when performing stochastic decodes from the 9 bit 4x4 latent space. Compression is from 98,304 to 144 bits (683x compression).

| Original | Stochastic Reconstructions |

Table 7: CelebA 128x128 reconstructions at different compression rates using HQA, with number of transmitted bits.

Figure 4: 'Free' samples obtained by exhaustively enumerating over all 256 codes from the 1x1 latent space of the trained MNIST HQA stack and decoding into pixel-space.

Figure 5: Rows show pairs of test images that have been encoded to the top of the HQA MNIST stack, interpolated across their codebook embeddings, quantized and then decoded.

Figure 6: Each row displays the diversity of stochastic decoding for a different held out MNIST image. First column is the original, then 14 stochastic decodes, and then final column is 14 averaged decodes. Class switching behaviour is displayed due to the high compression factor with a 1x1 latent bottleneck.

Figure 7: Samples generated by training a vanilla VAE on top of the learnt HQA 2x2 latent space and decoding first through the VAE then the HQA stack.

Table 8: Interpolations generated for each layer in HQA. The far left and right images are originals. Others are decoded from the interpolated encoder output $z_e$. Bottom row (HQA-1) has a compression ratio of 4, each subsequent layer compresses by 4 again until the final layer (HQA-5) results in an 8 bit 1x1 latent space. Lower layers exhibit blurriness and overlapping versions of originals but higher layers have increasingly dense support allowing realistic and coherent looking digits from anywhere in the latent space.

| System | Orig | Interpolation | Orig |
|--------|------|---------------|------|
| HQA-5 | 2 | 2 2 2 2 2 7 9 9 9 9 | 9 |
| HQA-4 | 2 | 2 2 2 2 3 3 9 9 9 9 | 9 |
| HQA-3 | 2 | 2 2 2 2 3 3 9 9 9 9 | 9 |
| HQA-2 | 2 | 2 2 2 2 2 3 9 9 9 9 | 9 |
| HQA-1 | 2 | 2 2 2 2 2 3 9 9 9 9 | 9 |

# B  Probabilistic VQ-VAE

## B.1  Motivation

In this section we outline the probabilistic model that motivates the HQA loss:

$$\mathcal{L} = -\log p(x|z=k) - \mathcal{H}[q(z|x)] + \mathbb{E}_{q(z|x)}||z_e(x) - e_z||_2^2 . \tag{5}$$

A desired property of the HQA, motivated in Section 4.4, is the non-deterministic posterior $q(z|x)$ defined over codebook space. For the HQA, this is defined as a softmax with logits equal to the negative squared Euclidean distances between the encoded points ($z_e(x)$) and codebook vectors ($e_k$):

$$q(z=k|x) \propto \exp -||z_e(x) - e_k||_2^2 . \tag{6}$$

This form of posterior occurs in a simple Gaussian Mixture Model (GMM), where they are referred to as *responsibilities*. In the GMM, the observed variables $x'$ are generated from possible sources $z' = 1, \ldots, N$. The responsibility of each source is then:

$$q(z'=k|x') \propto \exp -||x' - e_k||_2^2 . \tag{7}$$

This mirrors Equation 6 where the encoded point $z_e(x)$ is replaced by the observations $x'$. Therefore, in order to derive a Evidence LOwer Bound (ELBO) for our model, we use a small extension to the GMM that incorporates the encoder-decoder architecture.

## B.2  Probabilistic Model

(a) Gaussian Mixture Model Network

(b) A single layer of the HQA as a Bayesian Network

Figure 8: Contrasting the probabilistic model of a GMM and a single layer of the HQA Inference distributions are shown in red.

We introduce an additional latent variable $z_e$ into the standard GMM setup, so that the distribution $p(x|z)$ factorizes as:

$$p(x|z) = \underbrace{p(x|z_e)}_{\text{Decoder}}\underbrace{p(z_e|z)}_{\text{GMM}} . \qquad (8)$$

We contrast these two models in Figure 8. In this setup we treat $z_e$ as being generated from a GMM. $z_e$ is then fed through the decoder neural network.

To then infer a value for $z$ we first approximate the posterior $p(z_e|x)$ with a deterministic distribution on the output of the encoder neural network. To emphasize this in our analysis we refer to the output of the encoder as $z_e(x)$, whilst we refer to the latent variable as $z_e$. The final stage of inference to calculate $p(z|z_e)$ reduces to a simple GMM model with observed variables $x'$ in Equation 7 replaced with $z_e(x)$. This leads exactly to the posterior probabilities given in Equation 6. As $q(z_e|x)$ is deterministic we have that $q(z|z_e) = q(z|x)$ and so we use these expressions interchangeably.

This model is a Variational Autoencoder with a simple Mixture of Gaussians prior. In the prior, each Gaussian is assumed to be independent and have constant variance. Similar, more complex models are considered in Dilokthanakul et al. [7], Nalisnick et al. [26], Tomczak and Welling [37].

## B.3 Deriving the ELBO

Finally, as we have recovered the posterior probabilities we desire, we now derive the ELBO loss. For a general latent variable model with observation $x$ this is formulated as:

$$\mathcal{L}_{\text{ELBO}} = \mathbb{E}_{q(z|x)} \log p_\theta(x|z) - \text{KL}[q(z|x)||p(z)] \qquad (9)$$

where $q(z|x)$ is our approximate posterior distribution. However, in our case we have two latent variables, giving the loss:

$$\mathcal{L}_{\text{ELBO}} = \mathbb{E}_{q(z,z_e|x)} \log p_\theta(x|z, z_e) - \text{KL}[q(z, z_e|x)||p(z, z_e)] . \qquad (10)$$

We can then make use of the factorization in Equation 8 to rearrange this as:

$$\mathcal{L}_{\text{ELBO}} = \underbrace{\mathbb{E}_{q(z|z_e)q(z_e|x)} \log p_\theta(x|z_e)}_{\text{Reconstruction Loss}} - \underbrace{\text{KL}[q(z|z_e)q(z_e|x)||p(z)p(z_e|z)]}_{\text{KL to prior}} . \qquad (11)$$

We now consider each of these terms separately.

### B.3.1 Prior KL Loss

The Prior KL Loss is given by:

$$\mathcal{L}_{\text{prior}} = \text{KL}[q(z|z_e)q(z_e|x)||p(z)p(z_e|z)] . \qquad (12)$$

This factorizes into two separate KL terms

$$\mathcal{L}_{\text{prior}} = \text{KL}[q(z|x)||p(z)] + \mathbb{E}_{q(z|x)}\text{KL}[q(z_e|x)||p(z_e|z)] . \qquad (13)$$

As we define a uniform prior over mixture parameters $p(z)$, the first term becomes the entropy term $\mathcal{H}(q(z|x))$ as given in Equation 5. The next term is then:

$$\mathbb{E}_{q(z|x)}\text{KL}[q(z_e|x)||p(z_e|z)] = -\mathbb{E}_{q(z|x)} \log\left(e^{-||z_e(x)-e_z||_2^2}\right) = \mathbb{E}_{q(z|x)}||z_e(x) - e_z||_2^2 \qquad (14)$$

which is the final part of Equation 5. We omit two details: the constant terms and the factor of $0.5$ multiplied by the variance that usually occurs in the Gaussian density function as this is reweighted before training.

### B.3.2 Reconstruction Loss

The reconstruction loss is given by:

$$\mathcal{L}_{\text{recon}} = \mathbb{E}_{q(z|z_e)q(z_e|x)} \log p_\theta(x|z_e) = \mathbb{E}_{q(z_e|x)}\mathbb{E}_{q(z|z_e)} \log p_\theta(x|z_e) . \qquad (15)$$

In order to train with the quantized behaviour we require, we don't follow this calculation when calculating the reconstruction loss. Instead we sample from $q(z|z_e(x))$ and feed this back through the decoder. This modification gives

$$\mathcal{L}'_{\text{recon}} = \log p(x|z_e = k) \qquad (16)$$

where $k$ is sampled from $q(z|z_e(x))$. To clarify, whilst training, instead of using the encoded point $z_e$ as the input to the decoder, we feed the codebook vector sampled from the posterior $q(z|x)$.

### B.4 VQ-VAE as a limiting case

If we include a temperature parameter in our softmax posterior

$$q(z = k|x) \propto \exp\left(-\frac{1}{\tau}||z_e(x) - e_k||_2^2\right) \tag{17}$$

then as $\tau \to 0$, the posterior converges to a deterministic distribution:

$$q(z = k|x) = \begin{cases} 1 & \text{for } k = \text{argmin}_j ||z_e(x) - e_j||_2 \\ 0 & \text{otherwise} \end{cases} \tag{18}$$

This is precisely the posterior that arises in the VQ-VAE. In addition, the KL prior terms then become:

$$\mathcal{H}(q(z|x)) = 0 \tag{19}$$

$$\mathbb{E}_{q(z|x)}||z_e(x) - e_z||_2^2 = ||z_e(x) - e_k||_2^2 \tag{20}$$

If then stop gradient operators are applied to (20), the commitment and codebook loss from the VQ-VAE are recovered.

## C Architecture, training and hyper-parameters

### C.1 HQA

Each layer in the HQA stack is composed of an encoder, decoder and vector quantization layer. Encoders and decoders are feed forward networks composed of convolutional layers with 3x3 filters. Optional dilated convolutions are used in the decoder to increase the decoder's receptive field. Each code in the VQ layer codebook is represented by a 64 dimensional vector. The input $\hat{z}_e$ to layers 2 and above are normalized using running statistics, which was shown to stabilise training. A sigmoid activation is applied to the output of the decoder in the first layer.

The downsampling needed for compression is achieved through a strided convolution in the encoder and upsampling through nearest neighbour interpolation in the decoder. Each HQA layer is trained greedily with an MSE loss; gradients are only back-propagated through that single layer. For the first layer, the loss is taken between input pixels and decoder outputs, while all other layers calculate the loss between the input embedding $z_e$ and the predicted $\hat{z}_e$.

Optimization is performed using RAdam [20] with a learning rate of 4e-4 which is cosine annealed in the final third of training. Each layer was trained with distributed training across 8 Nvidia TITAN RTX's for CelebA, whilst MNIST was trained on a single TITAN X. During training, the Gumbel softmax temperature is linearly annealed to 0.01, with an initial temperature of 0.4 and 0.66 for CelebA and MNIST respectively.

Table 9: Hyper parameters of HQA network used for CelebA experiment

| | L1 | L2 | L3 | L4 | L5 | L6 | L7 |
|---|---|---|---|---|---|---|---|
| Input size | 64 | 64 | 32 | 16 | 8 | 4 | 2 |
| Batch size | 1024 | 1024 | 1024 | 1024 | 1024 | 1024 | 1024 |
| Encoder layers | 3 | 3 | 3 | 3 | 3 | 3 | 3 |
| Decoder layers | 6 | 6 | 6 | 6 | 6 | 6 | 6 |
| Encoder hidden units | 64 | 64 | 512 | 512 | 512 | 512 | 512 |
| Decoder hidden units | 64 | 64 | 512 | 512 | 512 | 512 | 512 |
| Codebook size | 512 | 512 | 512 | 512 | 512 | 512 | 512 |
| $\beta_e$ (entropy loss coefficient) | 5e-5 | 5e-5 | 5e-5 | 5e-5 | 5e-5 | 5e-5 | 5e-5 |
| $\beta_c$ (commitment loss coefficient) | 5e-5 | 5e-5 | 5e-5 | 5e-5 | 5e-5 | 5e-5 | 5e-5 |
| Training steps | 100k | 100k | 100k | 100k | 60k | 30k | 30k |
| Dropout | 0.0 | 0.0 | 0.0 | 0.5 | 0.5 | 0.5 | 0.5 |

### C.2 HAMs

The implemented HAMs architecture follows Fauw et al. [8]. Notably, it implements an MSE loss on pixels but all other layers use cross entropy for the reconstruction term. Separate commitment and

Table 10: Hyper parameters of HQA network used for MNIST experiment

|  | L1 | L2 | L3 | L4 | L5 |
|---|---|---|---|---|---|
| Input size | 32 | 16 | 8 | 4 | 2 |
| Batch size | 512 | 512 | 512 | 512 | 512 |
| Encoder layers | 3 | 3 | 3 | 3 | 3 |
| Decoder layers | 3 | 3 | 3 | 3 | 3 |
| Encoder hidden units | 16 | 16 | 32 | 48 | 80 |
| Decoder hidden units | 16 | 32 | 48 | 80 | 128 |
| Codebook size | 256 | 256 | 256 | 256 | 256 |
| $\beta_e$ (entropy loss coefficient) | 1e-3 | 1e-3 | 1e-3 | 1e-3 | 1e-3 |
| $\beta_c$ (commitment loss coefficient) | 1e-3 | 1e-3 | 1e-3 | 1e-3 | 1e-3 |
| Training steps | 18k | 18k | 18k | 18k | 18k |

codebook loss terms are also used. The codebook is not learnt directly, but updated via an online exponential moving average version of k-means. For the CelebA experiment a smaller batch sizes where used than the 1024 used for HQA. This is because we found training of HAMs to be very unstable if large batch sizes were used.

Table 11: Hyper parameters of HAMs network used for CelebA experiment

|  | L1 | L2 | L3 | L4 | L5 | L6 | L7 |
|---|---|---|---|---|---|---|---|
| Input size | 64 | 64 | 32 | 16 | 8 | 4 | 2 |
| Batch size | 32 | 64 | 64 | 64 | 64 | 64 | 64 |
| Encoder conv layers | 3 | 3 | 3 | 3 | 3 | 3 | 3 |
| Decoder conv layers | 3 | 3 | 3 | 3 | 3 | 3 | 3 |
| Encoder hidden units | 64 | 80 | 256 | 256 | 256 | 256 | 512 |
| Decoder hidden units | 64 | 80 | 512 | 512 | 512 | 512 | 512 |
| Encoder residual blocks | 2 | 2 | 2 | 3 | 3 | 2 | 1 |
| Decoder residual blocks | 2 | 2 | 2 | 3 | 3 | 2 | 1 |
| Codebook size | 512 | 512 | 512 | 512 | 512 | 512 | 512 |
| $\beta$ (commitment loss coefficient) | 1 | 50 | 50 | 50 | 50 | 50 | 10 |
| Learning rate | 4e-4 | 4e-4 | 4e-4 | 4e-4 | 1e-4 | 1e-4 | 1e-4 |
| Training steps | 250k | 300k | 50k | 50k | 50k | 50k | 25k |

Table 12: Hyper parameters of HAMs network used for MNIST experiment

|  | L1 | L2 | L3 | L4 | L5 |
|---|---|---|---|---|---|
| Input size | 32 | 16 | 8 | 4 | 2 |
| Batch size | 256 | 256 | 256 | 256 | 256 |
| Encoder conv layers | 3 | 3 | 3 | 3 | 3 |
| Decoder conv layers | 3 | 3 | 3 | 3 | 3 |
| Encoder hidden units | 16 | 16 | 32 | 48 | 80 |
| Decoder hidden units | 16 | 26 | 40 | 58 | 96 |
| Encoder residual blocks | 0 | 0 | 0 | 0 | 0 |
| Decoder residual blocks | 0 | 0 | 0 | 0 | 0 |
| Codebook size | 256 | 256 | 256 | 256 | 256 |
| $\beta$ (commitment loss coefficient) | 0.02 | 0.02 | 0.02 | 0.02 | 0.02 |
| Learning rate | 4e-4 | 4e-4 | 4e-4 | 4e-4 | 1e-4 |
| Training steps | 18k | 18k | 18k | 18k | 18k |

## C.3 VQ-VAE

The implemented VQ-VAE [39] architecture is comparable to HAMs, with the noticeable exception that there is no hierarchy. The same compression rates are achieved through downsampling multiple times. The entire network is trained end-to-end as a single layer, instead of greedily with local losses. The layers denoted in the table below refer VQ-VAE systems with equivalent compression factors to the same HQA and HAM layers. In all instances predictions are made in pixel space. The residual block implementation is based on the original VQ-VAE. As with HAMs, small batch sizes had to be used for the CelebA experiment as large batch sizes lead to instability.

Table 13: Hyper parameters of VQ-VAE network used for CelebA experiment

|  | L1 | L2 | L3 | L4 | L5 | L6 | L7 |
|---|---|---|---|---|---|---|---|
| Input size | 64 | 64 | 32 | 16 | 8 | 4 | 2 |
| Batch size | 32 | 64 | 64 | 64 | 64 | 64 | 64 |
| Encoder conv layers | 2 | 3 | 4 | 5 | 6 | 7 | 8 |
| Decoder conv layers | 3 | 4 | 5 | 6 | 7 | 8 | 9 |
| Encoder hidden units | 64 | 80 | 256 | 256 | 384 | 400 | 512 |
| Decoder hidden units | 64 | 80 | 256 | 512 | 512 | 512 | 512 |
| Encoder residual blocks | 2 | 3 | 4 | 4 | 4 | 4 | 2 |
| Decoder residual blocks | 2 | 3 | 4 | 4 | 4 | 4 | 2 |
| Codebook size | 512 | 512 | 512 | 512 | 512 | 512 | 512 |
| $\beta$ (commitment loss coefficient) | 0.05 | 0.25 | 0.25 | 0.25 | 0.25 | 0.25 | 0.25 |
| Learning rate | 4e-5 | 4e-5 | 4e-5 | 1e-4 | 1e-4 | 1e-4 | 1e-4 |
| Training steps | 250k | 250k | 250k | 150k | 150k | 150k | 50k |

Table 14: Hyper parameters of VQ-VAE network used for MNIST experiment

|  | L1 | L2 | L3 | L4 | L5 |
|---|---|---|---|---|---|
| Input size | 32 | 16 | 8 | 4 | 2 |
| Batch size | 512 | 512 | 512 | 512 | 512 |
| Encoder conv layers | 2 | 3 | 4 | 5 | 6 |
| Decoder conv layers | 3 | 4 | 5 | 6 | 7 |
| Encoder hidden units | 22 | 40 | 50 | 62 | 78 |
| Decoder hidden units | 16 | 18 | 20 | 22 | 22 |
| Encoder residual blocks | 0 | 0 | 0 | 0 | 0 |
| Decoder residual blocks | 0 | 0 | 0 | 0 | 0 |
| Codebook size | 256 | 256 | 256 | 256 | 256 |
| $\beta$ (commitment loss coefficient) | 0.125 | 0.125 | 0.125 | 0.125 | 0.125 |
| Learning rate | 4e-4 | 4e-4 | 4e-4 | 4e-4 | 4e-4 |
| Training steps | 18k | 18k | 18k | 18k | 18k |

## C.4  Codebook Resetting

During training, the total number of times that $z_e$ is quantized to each code is accumulated over 20 batches. After these 20 batches, the most and least used code, $e_m$ and $e_l$ respectively, are found. If the usage of $e_l$ is less than 3% than that of $e_m$, the position of $e_l$ is reset such that $e_l := e_m + \epsilon$ where $\epsilon \sim N(0, 0.01)$. This scheme is activate for the first 75% of training.

## D Algorithm description

---

**Algorithm 2** HQA Training

---

1: $e$: codebook embeddings, $e_k$: embdding for code $k$, $N$: number of codes in each layer
2: $L$: number of layers in stack
3: $\theta_i \leftarrow$ Initialize network parameters for encoders ($Encoder_i$) and decoders ($Decoder_i$) $\forall i \in L$
4: **for** $l$ in $1, \ldots, L$ **do**                                                              ▷ Train each layer greedily
5:     $\tau \leftarrow 0.4$                                                                    ▷ Set initial codebook temperature
6:     **while** not converged **do**
7:         $X \leftarrow$ Random minibatch
8:         **if** $l = 1$ **then**
9:             $\boldsymbol{z_{e-lower}} \leftarrow$ X
10:        **else**
11:            $\boldsymbol{z_{e-lower}} \leftarrow Encoder_{0..l-1}(X)$ ▷ Encode up through pre-trained lower layers - no quantization
12:        **end if**
13:        $\boldsymbol{z_e} \leftarrow Encoder_l(\boldsymbol{z_{e-lower}})$
14:        $p(k|\boldsymbol{z_e}) = \exp\left(-\frac{1}{2}||\boldsymbol{z_e} - \boldsymbol{e_k}||_2^2\right) / \sum_{i=1}^{N} \exp\left(-\frac{1}{2}||\boldsymbol{z_e} - \boldsymbol{e_i}||_2^2\right)$     ▷ Distribution over codes
15:        softonehot $\sim$ RelaxedCategorical$(\tau, p(k|\boldsymbol{z_e}))$          ▷ Reparameterized Gumbel-softmax sample
16:        $\boldsymbol{z_{q-soft}} \leftarrow$ softonehot $* \boldsymbol{e}$                          ▷ Soft quantized codebook lookup
17:        $\hat{\boldsymbol{z}}_{\boldsymbol{e-lower}} \leftarrow Decoder_l(\boldsymbol{z_{q-soft}})$
18:        $\mathcal{L}'_{recon} = (\hat{\boldsymbol{z}}_{\boldsymbol{e-lower}} - \boldsymbol{z_{e-lower}})^2$
19:        $\mathcal{L}_{entropy} = \sum_k p(k|\boldsymbol{z_e}) \log p(k|\boldsymbol{z_e})$
20:        $\mathcal{L}_{commit} = \sum_k p(k|\boldsymbol{z_e})||\boldsymbol{z_e} - e_k||_2^2$
21:        $\theta_i \leftarrow \theta_i - \eta \nabla_{\theta_i}(\mathcal{L}'_{recon} + \beta_e\mathcal{L}_{entropy} + \beta_c\mathcal{L}_{commit}))$
22:        $\tau \leftarrow anneal(\tau)$                                                          ▷ Anneal linearly
23:    **end while**
24: **end for**

---

**Algorithm 3** HQA Reconstruction

---

1: $e$: codebook embeddings
2: $L$: number of layers in stack
3: Trained encoders ($Encoder_i$) and decoders ($Decoder_i$) $\forall i \in L$
4: $x$: Datapoint to reconstruct
5: $\boldsymbol{z_e} \leftarrow Encoder_{0..l}(x)$
6: **for** $l$ in $L, \ldots, 1$ **do**
7:     $p(k|\boldsymbol{z_e}) = \exp\left(-\frac{1}{2}||\boldsymbol{z_e} - \boldsymbol{e_k}||_2^2\right) / \sum_{i=1}^{N} \exp\left(-\frac{1}{2}||\boldsymbol{z_e} - \boldsymbol{e_i}||_2^2\right)$ ▷ Distribution over codes
8:     onehot $\sim p(k|\boldsymbol{z_e})$
9:     $\boldsymbol{z_q} \leftarrow$ onehot $* \boldsymbol{e}$                                      ▷ Hard-quantized codebook lookup
10:    $\boldsymbol{z_e} \leftarrow Decoder_l(\boldsymbol{z_q})$
11: **end for**
12: **return** $\boldsymbol{z_e}$

---

Note that for hard reconstructions at fixed rates, we do not necessarily need to perform hard-quantized codebook lookups except on the very top codebook. For simplicity, and to provide a single hierarchy where each layer can provide compression at a fixed rate, we anneal the temperature close to zero and at test time always perform hard quantization operations at each layer as outlined in Algorithm 3.