[Reviews · NeurIPS 2020]

Review 1

Summary and Contributions: Hierarchies of VQ-VAEs (e.g. VQ-VAE-2 and HAM) can be used for lossy compression at lower rates than simple VQ-VAEs. The authors propose modifications to current methods in this area, leading to higher quality reconstructions at very low rates without using slow autoregressive components. This includes a novel objective for training hierarchies of VQ-VAEs with stochastic posterior distributions.

Strengths: Although in my opinion not grounded and developed well enough, the argument in favor of a stochastic posterior (with each VQ-VAE layer learning to reconstruct the full distribution of the layer below, rather than a sample) is interesting and definitely worth exploring. Empirical results seem to be relatively impressive (at least for CelebA). This work is part of an active research area that is interesting e.g. for the compression and (deep) generative modeling communities. It is therefore relevant for NeurIPS.

Weaknesses: I believe the contributions might be a bit overblown. In particular, in the first bullet point the authors mention an "analysis as to why probabilistic quantized hierarchies are particularly well-suited to optimising the perception-rate tradeoff when performing extreme lossy compression", but there doesn't seem to be a thorough and conclusive analysis on this matter. Points (2, 3) and (4, 5) are in my opinion spread too thin and should be combined. I find the model description somewhat lacking in clarity. I think everything would be much more understandable with a simple, but more structured, exposition (with equations) of the model, and possibly also of VQ-VAE, as this work heavily builds on it. Section 4 is in my opinion quite unclear. First, details of HQA are mentioned before it is introduced. Then, the toy experiment is only very briefly explained, and definitely needs more details (which are not in the Appendix as far as I can see). It is hard to assess the correctness and significance of the argument put forward by the authors, without a thorough understanding of the setup. Third, the structure and writing of this section is confusing, especially Section 4.3. Finally, I don't get how it's fair to compare the toy 2-layer HQA with a simple VQ-VAE with 2-code latents. The authors ask whether it's possible to model the toy density correctly by using 2 codes only, but then solve the problem by stacking 2 VQ-VAEs, of which the bottom one has in fact 4 codes. I'm sure this is a misunderstanding on my part, but I think it means that the argument is not explained effectively enough. The empirical evaluation could be more extensive, for example regarding ablation studies, and other widely used datasets such as ImageNet or FFHQ. The concept of rate-perception tradeoff, and its relation to the rate-distortion tradeoff, could be explained in more detail and briefly discussed in the experimental section. There is significant overlap with previous work, which raises questions about the potential influence of this work on the community, especially since most of the novelty seems to come from introducing stochasticity. This is in general a nice idea, although the novelty is limited. But I believe the paper needs restructuring, a clearer exposition of arguments and methods, and maybe, if possible, even an expanded experimental section. === EDIT after rebuttal: Thanks for the clarifications. I'm raising my score, but I still think this work is not particularly novel and, perhaps even more importantly, the ideas&model should be presented more clearly. In my opinion this would allow this paper to have a broader impact than just the learned image compression community. Regarding additional experiments: that would of course be a plus, but it's not my main concern.

Correctness: Claims, method, and empirical methodology seem correct.

Clarity: The paper is generally well written. My clarity concerns are related to the technical presentation only.

Relation to Prior Work: Closely related work is discussed in enough depth. However, works that are more loosely related to this are not mentioned. For example, there is a very rich literature on hierarchical VAEs, and on VAEs with discrete latent variables, and it would be interesting to read arguments for/against using these other approaches.

Reproducibility: Yes

Additional Feedback:


Review 2

Summary and Contributions: The paper expands on prior work on vector-quantized VAEs (VQVAE) and hierarchical autoregressive image models (De Fauw, 2019) by presenting a new compression scheme called Hierarchical Quantized Autoencoders (HQA) with a novel loss objective in comparison to VQ-VAEs. The proposed model introduces a hierarchical latent structure and replaces the deterministic posterior of VQ-VAE with Gumbel Softmax relaxation. It can produce high perceptual quality despite high compression and without relying on autoregressive decoders used by VQVAE and VQVAE2. In the low-rate lossy compression context, the method outperforms the baselines with FID and downstream classification task in MNIST, and visually in Celeba 64x64.

Strengths: The formulation as a whole is novel and seems theoretically solid. The significance of the work seems most obvious in the context of very high compression rate. In CelebA 64x64, the quality of reconstructions degrades gracefully when compression increases, unlike with the other models (as seen in Table 2). The proposed hierarchical latent approach seems sensible and could be more robust and less intensive at evaluation-time than the approach taken in VQ-VAE, especially as it does not require an autoregressive decoder. These ideas seem relevant in the continuing research for improving the vector quantized generative models.

Weaknesses: My biggest concern is about giving the primary (non-ablation) results only on a single dataset (CelebA) and the use of only low-resolution images (32x32 and 64x64) across the experiments. VQ-VAE, VQ-VAE2, and HAM all present results also on 256x256 resolution. While it is plausible that the proposed method scales gracefully to the larger resolutions, in generative models this cannot be taken for granted, as it can e.g. lead to highly unstable behaviour or infeasible computational complexity (in absence of any discussion to the contrary by the authors). Because the closest prior work does show those higher resolutions, this omission makes the paper feel incomplete. Although the performance of the model in the context of very high compression (i.e. low rate) appears good in 64x64 images, it is not obvious how these benefits scale to the larger resolutions. While the paper emphasizes the compression view of generative modelling, it would be useful to see random sampling performance and FIDs also for random samples, not only for reconstructions. In absence of that, it is harder to put the paper fully into the context of previous models, which nearly always provide these figures. (To be clear, I do not find it a problem if the FIDs for this model are not good, but it is a fairly simple thing to measure for the sake of completeness.) Due to the resolution limitations and scarce details about the computational requirements, I cannot honestly judge whether the proposed method is as relevant as the authors imply. EDIT after rebuttal: In the light of the rebuttal and the discussions with other reviewers, I concede that the resolution question, though highly relevant, does not have to be a show-stopper. Hence I have upgraded my score.

Correctness: The theoretical exposition seems correct, but the empirical justification for the claims, though acceptable as such, seems too limited. In this reviewer's opinion, a single dataset used only up to 64x64 resolution is not enough for the primary experiments in this context. Given that FID is famously dependent on sample size, I find the authors should definitely mention it in the context of FID calculations. (However, I base my initial estimate on the assumption that the sample size was sufficient.) The discussion of mode-covering in Sec 4 is interesting and seem to imply a fundamental improvement over the VQVAE approach. However, the claim (L220-222) that "the HQA reconstructions display higher perceptual quality than both VQ-VAE and HAMs at all compression rates" seems false, at least as far as Table 2 is concerned. The difference compression rates higher than 43x seem obvious, but I am hard-pressed to see differences at the ones before that (at the left). There could be a difference, but this is not obvious at the level of visual comparison. Moreover, I would not consider two sample faces sufficient for that kind of evaluation. E.g. HAMs seems to fail considerably for 683x and onwards, but it would be illustrative to see this analyzed for at least 10 random inputs for each model (e.g. in the supplement), if you seek to make a strong qualitative argument.

Clarity: The writing was mostly clear, but there are some parts of the paper that were somewhat confusing, especially when referring to 'Mode-covering' in Sec 4.3 and earlier. Minor: having section 3 only consist of subsection 3.1 is bad style.

Relation to Prior Work: The relationship to HAM could be made clearer. L96-98 sums it up, but this seems to me like the most relevant reference, and the relationship should be elaborated more.

Reproducibility: Yes

Additional Feedback: I was somewhat divided about this paper, since I find the ideas (and the experiment in Table 2) largely compelling, but I am disappointed to see the primary evaluation only on CelebA 64x64, and ablation only for MNIST. (MNIST evaluation might be acceptable for the ablation study, if there were more primary experiments on higher resolution datasets.) Therefore, I must reluctantly give lower score than I would like, and to recommend the authors to expand the paper by another high-resolution dataset. I assume some of the experiments here may be computationally intensive, though for the HQA model, the computational burden should be less than with the autoregressive decoders, correct? However, I see no mention of the computational requirements for the training and evaluation. Also, if HQA is considerably faster to evaluate than e.g. VQ-VAE, as one would expect, I believe you could emphasize this more. Note on reproducibility: The provided source code seems to support only the MNIST experiments. Will you be releasing the whole source code?


Review 3

Summary and Contributions: The paper improves over VQ-VAE by learning a hierarchy of discrete latent variables. Contrary to VQ-VAE they don't need to learn autoregressive priors over the latent to achieve good reconstructions even at low bitrates. The main contribution of the paper is to propose to use a stochastic posterior for the latent and a new probabilistic loss. They show experimentally that the visual quality of the reconstructions of their methods is superior to other discrete auto-encoders.

Strengths: The experiments are convincing, showing that the proposed approach enables better reconstruction according to rFID, especially at low bitrates. The proposed approach seems like a good step in the direction of learning low bitrates representation of images.

Weaknesses: The novelty of the approach is quite limited. The approach is close to VQ-VAE, the main difference being the stochastic posterior (already proposed in the literature as mentioned in the paper) and the probabilistic loss. It's not clear in the paper to which extent the probabilistic loss is important. The paper claim that it helps stabilize training on CelebA, but without any numbers or figure supporting this statement.

Correctness: The proposed approach although not supported by a strong theoretical grounding, is to some extent "close" to some ELBO objective as shown in the appendix of the paper. The empirical methodology is correct and supports the validity of the approach. However it would have been nice to have some confidence intervals about the results provided.

Clarity: The paper is overall well written and quite clear. However, I find section 4 not very informative.

Relation to Prior Work: The authors clearly discussed the related work, however the proposed approach is equivalent to fitting a hierarchical prior over the latent variables, in that sense it's related to autoregressive models that try to fit an autoregressive prior over the latent. I think this could be discussed a bit more.

Reproducibility: Yes

Additional Feedback: === Edit after rebuttal === I took into account the author response. I will maintain my score as I think the paper could still be improved but I think it meets the bar for acceptance. I encourage the author to clarify section 4 and the contributions in the next version and include results on higher resolution as mentioned by other reviewers.


Review 4

Summary and Contributions: This paper proposes to use a hierarchy of VQ-VAEs for lossy image compression. A stochastic posterior and adjusted training objective are introduced to better fit such a model to this task. Experiments show good performance at the challenging low bitrate regime.

Strengths: - The scheme proposed in this paper effectively tackles the challenging problem of maintaining realism and high-level semantic content in the scenario of (extreme) low-rate image compression. - Using a VQ-VAE stack jointly with probabilistic quantization seems to be an elegant method for remedying the mode covering / mode dropping behavior which is often present in image compression / reconstruction tasks. The authors present a very good explanation of how a probabilistic quantized hierarchy can alleviate these problems (section 4). - The proposed method performs better than the baselines both on common metrics and in qualitative visual comparisons. In particular, this model excels at the low bitrate regime, where obtaining realistic reconstructions is very challenging. - This paper can have a significant impact on the sub-community working on these tasks.

Weaknesses: - I think comparisons to prior work on generative lossy image compression (not only to prior work on hierarchical stacks of VQ-VAEs) are missing. In particular, a comparison to ref [39] (for which online code is available) could be interesting. - This paper builds on prior work which proposed similar concepts. The authors indeed discuss these works and the added contribution well, yet this somewhat limits the novelty.

Correctness: Yes.

Clarity: Very well written.

Relation to Prior Work: Yes.

Reproducibility: Yes

Additional Feedback: ===== Edit after rebuttal ===== After reading all the reviews and the author feedback, I think this is a very good paper and recommend it be accepted. The author feedback on the main points raised in the reviews is convincing. I would encourage the authors in any final revision: (a) To clarify section 4. (b) To include experiments at higher resolution.

[Author Response · NeurIPS 2020]

We thank all the reviewers for their feedback. We are encouraged by the praise received for both our motivation for quantized hierarchies (R2,R4) and empirical results (R1,R2,R3,R4). We are also pleased to see a recognition of significant impact for the generative compression sub-community (R4). All reviewers noted the effectiveness of HQA in the extreme low bitrate regime, without needing autoregressive components - this was a key contribution we wanted to emphasise. We will incorporate all feedback in a future revision but address key themes below.

**Praise & Criticism Of Section 4 (Lossy Compression Using Quantized Hierarchies)** A key contribution of our work is motivating why, under an information bottleneck, *quantized hierarchies with stochastic quantization* show a fundamentally different modelling behaviour from other likelihood-based systems. We believe we are the first to present this form of motivation. We were glad to see R4 recognise this section as being a "very good explanation" of how quantized hierarchies remedy common pathologies of likelihood-based image compression and R2 notes "a fundamental improvement over the VQ-VAE approach". However, as R2 comments, the "structure and writing of this section is confusing, especially Section 4.3". Noted - this section is important but the reviewers have convinced us that the exposition needs a significant rewrite. To summarize, we show how quantized hierarchies exchange mode-covering behaviour in the input space for mode-covering behaviour in the latent space. Stochastic quantization then allows for high quality reconstructions without sacrificing diversity.

R1 gives several excellent suggestions for how the clarity of this section can be improved, which we think will lead to a much stronger paper. R1: "I don't get how it's fair to compare the toy 2-layer HQA with a simple VQ-VAE with 2-code latents" - when comparing two compression schemes it is only the number of codes in the top layer that is controlled for as this ultimately determines the model's compression rate.

**Experiments** We are glad to see that all authors found our empirical results impressive. However, R1 and R2 would have found our experimental section more convincing if we included results on higher resolution images, such as FFHQ and ImageNet. We completely agree - the criticism is entirely valid. In particular, we thank R2 for their helpful analysis of the relationship between FID and image resolution. We are saddened to say that we simply don't have the resources to pursue these higher resolutions. VQ-VAE, VQ-VAE2 and HAMs were all developed by the same research lab that has access to several orders of magnitude more compute than us. We stress that it is common for well received work in generative modelling and generative compression to only quote on 64x64 images [1, 2, 3]. However, we agree that higher resolution evaluations would have significantly improved the paper - we will provide 128x128 reconstructions and make a best effort to obtain the compute budget needed to train a 256x256 system.

The other points raised regarding our experimental section are more easily addressed. As R2 notes, our experiments were designed around the generative compression setting, as opposed to generative modelling - we stand by this evaluation protocol and leave sampling for future work. R2 also notes that the two faces we show in Table 2 are not quite enough to convince them of a strong qualitative argument; we have plenty more comparisons which we will add into the appendix to solidify our qualitative argument. R4 notes a comparison to [3] would have been useful - noted - we actually did produce an internal comparison. We found the visual quality of HQA to be preferable but the FID scores of [3] to be superior. We believe this is in part due to [3] being adversarially trained. We made the decision not to include these results as we wanted to present a fair investigation comparing *likelihood-based* approaches, as we discuss briefly in Section 6.1. We will include these results in future supplementary material.

After including the above results, we believe our work will be much stronger and will sufficiently address the scepticism around our experimental analysis.

**Novelty Clarifications** R1 and R3 expressed concerns about the novelty of our work. This may be linked to the misgivings R1 and R3 have regarding Section 4. The analysis of quantized hierarchies in Section 4 is a key contribution of our work. As we mention above, we strongly believe that an improved exposition in Section 4 will lead to both R1 and R3 deeming our paper more novel. We also wish to highlight the strength of the HQA scheme lies in our novel prediction target, namely reconstructing $z_e$ using MSE and not $z_q$ using cross-entropy (as done in HAMs). This small change has **dramatic** consequences, as shown by both Table 2 and Row 5 in Table 3. We believe this novel prediction target, and scheme taken as a whole, have large implications for those working on quantized hierarchies and set our work apart from others. (We would also wish to highlight to R3 that the novel prediction target is separate from the probabilistic commitment and codebook losses, which may address some of their confusion. We will clarify this in a future revision). We also appreciate R1's feedback on the introduction of our contributions and will rewrite according to their suggestions.

[1] A. Radford, L. Metz, and S. Chintala. Unsupervised Representation Learning with Deep Convolutional Generative Adversarial Networks.

[2] A. Genevay, G. Peyre, and M. Cuturi. Learning Generative Models with Sinkhorn Divergences.

[3] M. Tschannen, E. Agustsson, and M. Lucic. Deep generative models for distribution-preserving lossy compression.


[Meta-Review · NeurIPS 2020]

Post-rebuttal, 3 out of 4 reviewers vote for acceptance while R1 esteems it is marginally below threshold. All 4 reviewers appreciated the convincing results in low-bitcount regimes. The two main points debated in the discussion phase concerned: a) whether there was sufficient novelty in the work w.r.t. VQ-VAE related prior-works. For this, R4 convincingly argued that the paper significantly contributes to the learned compression sub-field. W.r.t. VQ-VAE, taking into account the author response and following reviewer discussion, the AC agrees that there are original aspects in the proposed algorithm that make it markedly different, with reasonable justification and ablation. The AC judges that the new outlook proposed will be useful for the research community to ponder. b) whether the rather small scale experiments (in particular the 64x64 resolution) are sufficient by current standards. -> for this authors said in their reply that they would add results of experiments with higher resolution. A remaining criticism of R1 is that the ideas and model should be presented more clearly. Given its judged valuable contribution to the learned compression sub-field, and the original aspects of the approach being worth pondering by the community interested in learning quantized representations, the AC recommends acceptance. But in accordance with the reviewes requests that the final revision: (a) clarifies section 4 and 5. (b) includes experiments at higher resolution as promised by the authors.